# AI in Healthcare for Resource Limited Settings: An Exploration and Ethical Evaluation

Rune Chi Zhao*
runechi.zhao@anu.edu.au
Australian National University

Xiuyuan Yuan†
xiuyuan.yuan@anu.edu.au
Australian National University

## Abstract

Advances in artificial intelligence (AI) hold great promise to transform healthcare in resource-limited settings (RLS). However, due to challenges such as shortages in healthcare professionals, data scarcity, and inadequate regulatory frameworks, RLS are left especially vulnerable to AI's potential risks and ethical violations. Thus, despite rising expectations, substantial gaps remain in our understanding of how to responsibly integrate AI into the healthcare systems of RLS. In response, this review critically examines AI healthcare applications in RLS, with the intention of promoting ethical, transparent, and secure principles for future implementations. We first provide an exploration of the potential uses of AI in resource-limited healthcare and present four broad subfields: decision support, predictive analytics, telemedicine and digital health tools, and resource management. Taking an analytical approach, we illustrate both the potential benefits and hidden ethical pitfalls that may arise when implementing AI in contexts with limited human and financial resources. Drawing on both recent studies and original perspectives, we aim to provide an overview of major ethical concerns in an RLS context - including algorithmic bias, non-maleficence, privacy and security, autonomy, and transparency – as well as discussing additional ethical dilemmas rarely addressed in literature. Subsequently, we advocate for context-specific regulations and culturally sensitive frameworks, in addition to robust oversight and active local participation. Finally, we provide recommendations that aim to protect patient welfare, uphold autonomy, and promote equity—so that AI applications ultimately strengthen, rather than undermine, global efforts to reduce healthcare disparities.

## CCS Concepts

• **Computing methodologies** → **Artificial intelligence**; • **Social and professional topics** → **User characteristics**; • **Security and privacy** → **Human and societal aspects of security and privacy**; • **Applied computing** → **Life and medical sciences**;

## Keywords

Artificial Intelligence, Healthcare, Resource Limited Settings, Ethics, Privacy and Security, Low or Middle Income Countries, Algorithmic Bias, Autonomy, Transparency

**ACM Reference Format:**
Rune Chi Zhao and Xiuyuan Yuan. 2025. AI in Healthcare for Resource Limited Settings: An Exploration and Ethical Evaluation . In *Proceedings of (WWW '25, TIME '25[1])*. ACM, New York, NY, USA, 8 pages.

*Corresponding Author, Undergraduate student
†Undergraduate student

## 1 Introduction

Over the last several decades, artificial intelligence (AI) has evolved from a conceptual frontier to a tangible force shaping numerous fields, including healthcare. The transformative power of AI in healthcare has already been demonstrated in a range of contexts, including diagnostics, medical imaging, surgical robotics, and more, contributing to the ongoing global digital transformation of health services [2]. As computing power, data availability, and algorithmic sophistication advance, AI is becoming an increasingly indispensable ally in the healthcare industry, empowering professionals and providing valuable insights [3]. However, despite growing enthusiasm and expectations, significant gaps remain in our understanding of how to ethically and effectively deploy these advanced tools in resource-limited settings [4, 5].

Resource-limited settings (RLS) refer to regions or communities where access to essential healthcare services, infrastructure, and economic resources is significantly constrained [6, 7]. Typically encompassing low- and middle-income countries (LMICs) in regions such as Latin America, Asia, and Africa, but also including underserved rural areas in high-income countries, RLS are characterized by unstable economic growth, limited access to new technologies, and diminished research opportunities [3, 4, 7]. As a consequence, the quality of healthcare is significantly lowered, with shortages of healthcare professionals, lack of diagnostic tools, and higher disease burdens [3, 8, 9]. For example, to illustrate the vast difference between resource-limited and resource-rich settings, Nigeria has approximately 380 critical care nurses, whilst the USA, which has less than three times the population of Nigeria, has over 500,000 critical care nurses [8, 10, 11].

In recent years, the potential of AI to help "bridge the gap" in healthcare quality between RLS and resource-rich settings has been increasingly recognised, with new technologies developed in an attempt to diminish global disparities [9]. By assisting with diagnostic decisions, treatment plans, allocation of medical resources, and automating time-consuming tasks, AI can optimise healthcare systems and alleviate the workload of healthcare professionals, which is especially valuable in RLS.

However, the reality is that AI healthcare applications have mostly been deployed and then evaluated in high-income countries, with developments in RLS being nascent in comparison [2]. Furthermore, while the literature on AI in healthcare has rapidly expanded in recent years, the clear majority of studies are anchored in high-income settings where increased accessibility and robust infrastructures make it easier to deploy new technologies [5, 12, 13]. In order to correctly deploy AI tools in RLS, it is imperative to first conduct evaluations of how AI can be responsibly and effectively integrated. Thus, this lack of ethical evaluation in RLS needs to be urgently addressed, as cultural, socioeconomic, and infrastructural

realities differ substantially from those in wealthier environments, in addition to persistent material and systemic constraints [3, 5].

In existing studies based in RLS, discussions mostly focus on feasibility, performance, and impact factor, with less attention paid to ethical implications [3, 5]. Concerns over bias, privacy, autonomy, and culturally appropriate design are vital yet insufficiently addressed in these unique settings, leaving a critical need for guidance that integrates not just technical insights but also ethical principles and community-centric perspectives.

Thus, this review evaluates the promise of AI-driven healthcare interventions in resource-limited settings through a critical and ethical lens. In doing so, we hope to look beyond technical feasibility, calling for a holistic understanding that aligns AI deployments with ethical standards, respects human rights, and ultimately improves health outcomes in some of the world's most vulnerable regions.

## 2 Method

In undertaking this review, the primary goal was to gather a broad yet representative set of studies that illustrate the current state of AI healthcare applications in RLS and uncover ethical concerns specific to such environments. To achieve this, a survey-natured narrative review approach was adopted, collating a diverse range of studies from 42 distinct sources to explore the current landscape. Given the still-emerging nature of AI applications in RLS, a narrative review allowed for flexibility, enabling the inclusion of diverse source types and study designs, rather than constraining the analysis to a single study type or purely quantitative metrics. This approach highlights thematic and contextual insights, complementing past systematic reviews and offering an integrative perspective, which was deemed most suitable given the evolving nature of AI and multilayered ethical challenges in RLS.

Databases consulted included PubMed and Google Scholar, using combinations of search terms such as "artificial intelligence," "healthcare," "global health," "low- and middle-income countries," "resource-limited settings," "ethics," "bias," and "fairness". The time frame was restricted primarily to publications from 2010 onward to capture the relatively recent, rapid developments in AI. Older foundational works were retained only if they held continued relevance, especially for ethical and conceptual underpinnings.

Regarding inclusion and exclusion criteria, studies were selected if they reported original research evaluating AI applications in healthcare, addressed ethical issues in AI-driven healthcare, discussed specific frameworks for equitable implementation, or discussed barriers and facilitators to AI adoption relevant to RLS. Studies or commentary pieces solely focusing on AI implementations in high-income contexts without any comparative or transferrable insights to RLS were excluded. Where possible, references that provided case studies or practical examples of AI deployment (e.g., telemedicine programs, diagnostic models) in specific RLS regions were prioritised.

## 3 AI Healthcare Applications for RLS

At its core, AI involves designing computational systems that can perform tasks typically associated with human intelligence, such as learning from experience and making reasoned decisions [14]. Two overarching categories are frequently highlighted: artificial

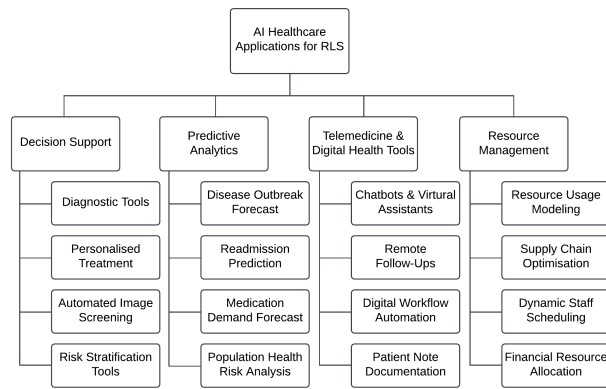

**Figure 1: Overview of common AI healthcare applications for RLS**

general intelligence (AGI) and artificial narrow intelligence (ANI) [2, 15]. AGI aims to replicate the full spectrum of human cognitive abilities, whilst ANI focuses on excelling in specific, well-defined tasks [15]. Most current healthcare applications fall under the ANI umbrella, employing subfields like machine learning, natural language processing, expert systems, computer vision, and automated planning to tackle different medical challenges [2, 3]. Using the sources gathered using the methods described above, we examined different medical applications of AI in RLS of which we sorted into 4 broad categories, as seen in Figure 1. However, it should be noted that overlap between these categories is common, and certain applications may belong to more than one category.

### 3.1 Decision Support: Diagnostics and Treatment Plans

Decision support systems, commonly referred to Clinical Decision Support Systems (CDSS) or Computerized Decision Support (CDS), assist clinicians by providing evidence-based guidance, helping them diagnose conditions, create treatment plans, monitor patients, and manage overall patient care. AI-assisted decision-making is powered by models which employ a diverse range of algorithms, including supervised learning, neural networks, and ensemble methods, to identify patterns from different datasets [16, 17]. As an output, the model may provide a probability assessment upon which clinicians can make quick and informed decisions, as well as treatment and other action recommendations depending on the input material. Currently, most experts agree that such algorithms themselves are not reliable enough to act as the final decision-maker, which is reflected in global regulatory frameworks [16]. Thus, CDSS are primarily employed as screening tools and diagnostic or treatment aids, helping analyse patient data such as medical images, lab results, and patient images [17].

In RLS, decision support systems can be an incredibly valuable tool as they can compensate for shortages in specialists, up-to-date guidelines, and advanced diagnostic equipment. By automating parts of the decision process, AI reduces clinical workload and

improves consistency of care, even when human expertise is limited, ultimately bridging gaps in accuracy and efficiency. In Peru, a computer-aided tuberculosis (TB) diagnosis system was recently trialled, where a deep learning algorithm was used to identify manifestations of TB based on patient information and chest X-ray photographs, as uploaded by nurses [18]. Due to the underfunded public healthcare sector and severe understaffing of healthcare professionals, TB diagnosis in Peru is an extremely slow process, causing fatal delays in treatment [18]. Thus, this AI-assisted system hopes to improve the speed and reliability of chest X-ray readings [19]. Another decision support system has been trialled in Africa, where cancer is becoming a growing public health challenge due to a severe shortage of oncologists. Gukiza, launched by Hurone AI, is a remote patient monitoring system which hopes to support clinical decision-making by providing treatment insights from patient reports of side-effects and symptoms [20].

## 3.2 Predictive Analytics

By analysing historical and real-time data, AI is able to predict future health trends, enabling healthcare professionals to take proactive interventions. Machine learning and deep learning algorithms can be applied to large datasets – including epidemiological records, environmental data, and electronic health records (EHRs) – which enable AI to identify patterns and correlations not easily discernible otherwise [21]. In RLS, predictive analytics can help predict disease outbreaks, identify patients at high risk of complications, and forecast medication demand [21, 22]. AI-driven epidemiology predictions are especially important for RLS, as studies have shown that LMICs and other RLS bear a disproportionately higher burden of both noncommunicable and infectious diseases, due to factors such as poor sanitation, less vaccination, poverty, and malnutrition [23]. Effective epidemiological surveillance is vital for RLS, as it allows public health authorities to implement interventions before the infection spread overwhelms their resources and capabilities. Traditional epidemiological tracking is labour-intensive and inefficient due to its reliance on manual data collection and reporting, highlighting the need for AI-driven predictive models in RLS [23]. Currently, models have been used for predicting disease outbreaks, identifying populations at risk of low vaccination uptake, and forecasting seasonal infection trends [23, 24].

In South Africa, researchers have developed fuzzy logic-based AI models that can provide early warnings for cholera outbreaks by analysing environmental and biophysical parameters like water quality, as well as epidemiological data [25]. Cholera is a highly infectious disease, with outbreaks in rural and impoverished communities being especially hard to control [25]. Therefore, accurate forecasting of outbreak risk potential is crucial for preventative measures, illustrating the value of predictive analytics in communities lacking robust healthcare infrastructure.

In addition to larger-scale applications like epidemiology, AI can also be used in conjunction with wearable devices that monitor vitals in real-time to provide predictive alerts [21]. Thus, healthcare professionals can detect potential issues before they occur, leading to faster interventions and better patient management [21]. One example is an AI model that can identify infants at risk of late-onset sepsis through non-invasive monitoring, developed by researchers

in the Netherlands [26]. This predictive model is especially valuable for RLS, with LMICs accounting for 99% of global neonatal mortality, highlighting how AI applications can be used to reduce global disparities in healthcare [27].

## 3.3 Telemedicine and Digital Health Tools

Telemedicine leverages communication technologies and AI subfields like natural language processing (NLP) to deliver healthcare remotely [28]. Recently, telemedicine and virtual care have become increasingly common as an alternative to in-person care. This surge in popularity can attributed to complementary advances in digital technologies like mobile health and cloud computing [2]. Furthermore, increased mobile phone penetration in RLS and substantial investments in digital health have provided many resource-limited and developing areas with the necessary basics to initiate AI applications [2].

In RLS, telemedicine can be an extremely cost-effective and efficient solution for addressing both geographical and systematic barriers, such as distance and low doctor-to-patient ratios. AI-enhanced telemedicine platforms can facilitate virtual consultations, preliminary assessments, and follow-up care [28]. Chatbots and virtual assistants, powered by NLP, can also converse with patients regarding their symptoms, guiding them on whether to seek in-person care, manage conditions at home, or connect with a clinician remotely [4, 28, 29]. Thus, telemedicine benefits the healthcare system by easing pressure on understaffed facilities, whilst also benefiting patients by reducing waiting times, travel costs, and travel time [28].

In China, self-diagnosis chatbots have been widely deployed, with popular chatbots attracting over hundreds of thousands of users [4, 29]. One notable example is DoctorBot, an AI-driven, mobile-based platform for medical consultations [29]. Users can explain their health concerns to DoctorBot via text or voice, with DoctorBot providing diagnoses and personalised medical advice, such as treatment options or medication and diet suggestions [29]. DoctorBot was created to address the rising demand for healthcare services and the imbalanced distribution of resources, which together caused access to medical advice for individuals in rural regions to become insufficient [29]. A large-scale trial of DoctorBot was conducted in 2021, which found a majority of user feedback was positive, demonstrating the potential of chatbots in RLS [30].

In addition to telemedicine, NLP can also be used in other digital health applications that help to streamline tasks and improve the efficiency of healthcare professionals. In RLS, NLP can provide automated documentation of verbal patient notes, allowing doctors to expend more time and energy on direct patient care [3]. NLP can also help manage medical records and extract essential information from complex documents like clinical notes and lab reports, improving decision-making and reducing errors in high-demand settings [3].

## 3.4 Resource Management

AI can be used to derive insights for resource management, helping forecast supply needs, optimize the distribution of medical supplies, and enhance logistics in healthcare systems [31]. Due to the deeply

rooted issue of health inequality, there is a severely troubling difference in resource allocation between resource-rich and poor areas [32]. Consequently, in RLS, ensuring an uninterrupted flow of essential medicines, vaccines, and equipment is often difficult due to unreliable supply chains and poor inventory management [31]. AI applications can help RLS maximise their resources, optimising effectiveness and efficiency.

By analysing past data, such as historical consumption data, disease incidence patterns, and seasonal variations, AI algorithms can anticipate demand, including shortages and surpluses, and provide suitable recommendations for resource allocation. This can help hospitals and clinics maintain optimal stock levels, reducing waste whilst also ensuring the availability of essential goods [33]. For example, researchers have used an artificial neural network to build a financial resource allocation model for public health in Brazil, which intakes information such as proportion of elderly people, sanitation, and income to help allocate funding [31, 34].

Furthermore, AI-driven resource allocation is especially useful in emergency situations like pandemics, as fast decisions need to be made in a rapidly changing environment [31]. For example, during the COVID-19 pandemic, AI tools were used in many RLS to help allocate resources like medicine, hospital beds, and doctors, demonstrating their value in optimising the efficiency of healthcare systems [31].

## 4 Ethical Evaluation

Whilst the benefits of integrating AI into resource-limited healthcare systems are clear, it is critical that the ethical considerations of implementing new technologies are not ignored. In settings already marked by fragile infrastructure and limited regulatory oversight, integrating AI without careful ethical consideration could exacerbate existing inequalities and vulnerabilities, rather than alleviating them. Thus, as the use of AI-based healthcare solutions expands further into RLS, conducting a thorough and RLS-specific ethical evaluation becomes not just advisable, but essential.

Currently, the main ethical considerations of implementing AI applications in healthcare can be summarised into 5 areas: bias and fairness, non-maleficence, privacy and security, autonomy, and transparency, as seen in Figure 2 [2, 35]. In the following sections, we discuss these considerations in the context of RLS, which introduces an additional layer of complexity, with the aim of providing both an overview of current literature and unique perspectives. Whilst we acknowledge that ongoing developments in AGI may introduce additional ethical complexities, our analysis is primarily focuses on narrower AI applications, as they are more relevant to the current healthcare landscape in RLS.

### 4.1 Bias and Fairness

Bias is a major concern for AI applications in healthcare, especially for RLS due to disparities in data availability and reduced representation. The two main sources of bias in AI are data bias and algorithmic bias, which come from the training data and inherent algorithmic mechanisms respectively [36]. The datasets that models are trained upon are just as important as the AI algorithms themselves, as they are the basis from which the AI will learn patterns and derive insights. Therefore, if AI models are trained on datasets

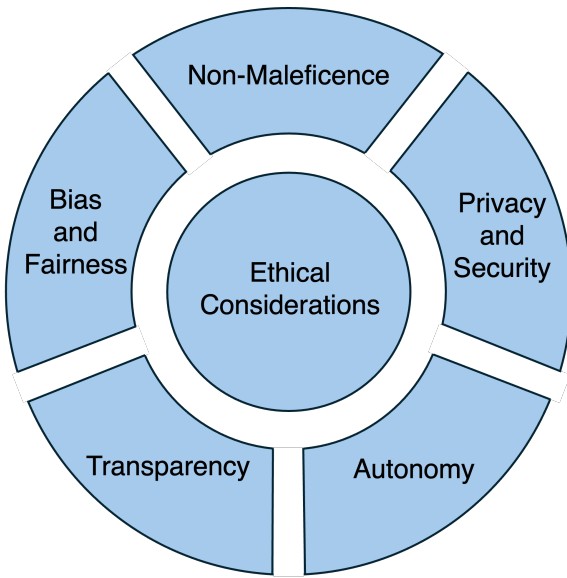

**Figure 2: Key ethical considerations discussed regarding to AI healthcare applications in RLS**

that reflect existing inequalities or cultural assumptions, they may produce outputs that disadvantage certain groups, worsening global health disparities [5, 32]. As algorithms often rely on statistically prominent patterns rather than social and ethical considerations, they may reinforce stereotypes and exclude marginalized populations from equitable care, ultimately introducing discrimination into supposedly fair computational processes.

It is important to note that despite their reputation for being more objective than humans, AI algorithms are inherently political, as they are borne from the choices and beliefs of their creators, whether conscious or not [35]. For example, certain population datasets have been found to be heavily biased by the discriminatory overdiagnosis of schizophrenia in African Americans [37]. If these datasets are then used to train AI applications used in African populations, there could be a surge of inaccurate diagnoses like false positives. In RLS, overdiagnosis could quickly exhaust resources, leading to disastrous consequences. Thus, it is possible that AI could actually help perpetuate and even worsen existing disparities in RLS, rather than improving them [35].

Furthermore, there is a global shortage of high quality datasets from RLS due to fragmented health information systems and disorganised data management practices [7]. Therefore, datasets often have to be sourced from high-income contexts, meaning that the training data substantially differs from the actual deployment [5, 35]. Due to drastically different conditions and healthcare systems, datasets from high-income settings fail to capture local health dynamics and cannot accurately represent populations from RLS, especially the diverse range of minority groups [5]. Thus, the adoption of AI technologies that have been trained in different settings may result in contextual bias and unexpected consequences for the local communities.

The ethical issue of fairness is intrinsically connected with bias, as it refers to the equitable distribution of resources, opportunities,

and outcomes [36]. Regardless of social status, ethnicity, or gender, healthcare systems should provide all patients with the same quality of care. As stated by the World Medical Association's Geneva Declaration, no factors should influence a physician's duty to their patients [38]. Thus, the risk of bias and discrimination in AI-driven tools is a critical threat to fairness in healthcare systems.

## 4.2 Non-Maleficence

The "do no harm" principle is often considered the first rule of medicine, making non-maleficence a fundamental ethical principle in medical practice [13]. Thus, ensuring non-maleficence is a critical ethical concern when applying AI solutions to healthcare, especially as mistakes simultaneously affect large numbers of patients, amplifying the impact of each error [13]. In RLS, the potential for harm is further magnified by fragile health infrastructures, which can compound the risk of exploitative or unsafe AI implementations.

Inaccurate AI models can cause harm in many ways, with one example being the occurrence of misdiagnoses or overdiagnoses [39]. Misdiagnoses can result in false hope, uncertainty, and delayed care, whilst overdiagnoses can lead to unnecessary interventions and psychological distress, both of which undermine the principle of "do no harm" [40]. Furthermore, AIs developed in high income settings may recommend treatments that are locally unavailable or too expensive for patients from RLS, causing patients even more harm and distress as they cannot access such care.

Finally, if AI health care technologies in RLS are not properly governed and managed, there is a strong potential for misuse. For example, AI technologies have been used to deduce ethnicities in patients before - which is relevant for certain clinical cases - however this ability could also be used for racial profiling if placed in the wrong hands [13]. RLS often lack robust regulatory frameworks and policies regarding new technologies, making it easier for AI tools to be used by oppressive forces [13]. As a consequence, AI could possibly deepen pre-existing inequities and reinforce hierarchical social structures, causing even greater harm to vulnerable populations. While AI holds tremendous promise for enhancing healthcare in RLS, it also has the capacity to inflict harm if misused. Ensuring non-maleficence is therefore critical, as the ultimate impact of these powerful tools depends on the intentions of those who deploy them.

## 4.3 Privacy and Security

Ensuring the privacy and security of patient health data is another fundamental ethical obligation when deploying AI-driven healthcare tools, yet it remains an ongoing challenge in RLS [35, 41]. AI systems rely on large amounts of sensitive data, such as patient health records, laboratory reports, and even genomic information for certain applications [41]. Therefore, it is crucial that the personal information of patients is safeguarded, as this sensitive information could potentially be used to discriminate. However, in low-resource environments where governance, enforcement mechanisms, and digital literacy are often weak, the lack of robust policies and data stewardship practices makes it more difficult to ensure that sensitive health information is handled responsibly [35, 41]. Thus, AI applications that utilise private data in RLS may be more vulnerable to breaches and unauthorised access, putting both individual

and societal well-being at risk. Furthermore, current legislation for privacy and data protection often has ambiguous wordings and outdated definitions in regard to AI applications, rendering them insufficient at safeguarding data, especially from large corporations whose involvement in healthcare expands beyond traditional covered entities [42].

These vulnerabilities carry significant ethical implications, as any unauthorized disclosure or exploitation of personal health information not only erodes patient trust and autonomy but may also reinforce social inequities [35, 42]. If patients lack confidence in data protections or fear misuse, they may be less willing to share their sensitive medical information for the development of AI applications. Thus, it is crucial to address privacy and data security concerns for RLS, in order to promote protected and robust AI applications in healthcare.

## 4.4 Autonomy

Respecting patient autonomy is another fundamental principle in healthcare, as all individuals should have the right to make informed decisions about their own care [3]. The ethical consideration of human autonomy preservation has been one of the most frequency discussed in literature pertaining to AI in healthcare [35]. The introduction of AI-driven systems risk reducing the opportunity for patient input and shared decision-making, especially if models rely on generalised data instead of patient-specific details [35]. Shared decision-making is vital, as it enables patients to have a dynamic dialog with their doctor and make informed and well-advised decisions, that still consider their unique belief systems [40]. However, the concern is that AI decision making tools may not take different patient preferences and values into account, which is especially problematic for RLS, where patients may already have reduced opportunities for medical autonomy [40]. Patient outcomes are also strongly influenced by a patient's self-perceived quality of life, which AI models might not capture due to its subjective and qualitative nature [40]. Such tools may inadvertently push care in a more overbearing and rigid direction, especially when predictions about patient outcomes—derived from algorithmic models—lack the nuance needed to address individual patient beliefs, cultures, and personal circumstances [35]. Without adequate clinician training to ensure patient understanding, there is a risk that AI models will overshadow individual desires, inadvertently pressuring patients into accepting certain interventions. If these concerns are not adequately addressed, patients in RLS may even begin to avoid seeking medical care due to a fear of losing autonomy.

Autonomy also requires patients and surrogate decision-makers to have a sufficient understanding of the medical information, as outcomes are rarely binary and are strongly correlated to self-perceived quality of life [40]. However, AI applications may be less effective than trained clinicians in communicating and be more likely to bypass patient understanding for efficiency. This risk is amplified in RLS, as health and technological literacy is already disproportionately lower, meaning that the introduction of AI may exacerbate pre-existing challenges in patient understanding [43, 44]. Furthermore, due to AI's complexities and common mis-understandings, clinicians are vital in helping interpret outputs, such as probability-based prognostications [40]. However, shortages of clinicians in

RLS may lead to patients having to confront AI recommendations without the necessary support.

## 4.5 Transparency

The transparency of AI algorithms is a key principle that is inherently connected with many other ethical considerations. Transparency refers to the ability to understand how AI systems make decisions and generate specific outputs from given inputs [40, 45]. Without transparency, there is an overall loss of trust and predictability, and so it is important to understand the logic behind different AI algorithms [35]. Currently, AI algorithms range from simple and easily explainable structures – such as decision trees and logistic regression – to more complex and sophisticated models – such as transformers and deep neural networks [46]. It is these more complicated algorithms that can become "black boxes", meaning that humans cannot understand their decision-making process [45]. In healthcare, the black box problem leads to a lack of clarity and certainty, making it difficult for both professionals and patients to fully trust AI recommendations [35]. The issue of potentially opaque models is especially concerning for RLS, as healthcare systems have fewer tools and opportunities to validate algorithmic outputs.

Furthermore, limited transparency undermines patient autonomy as individuals are not able to gain insights into why a system has proposed a particular course of action—especially when these recommendations may conflict with their personal preferences or cultural values [35]. In RLS, patients already face barriers like low digital and medical literacy, leading to higher risks of losing autonomy [44]. In addition, if AI recommendations cannot be easily explained, patients may be hesitant to rely on these systems, leading to mistrust of otherwise beneficial technologies. Ultimately, a balance between computational power and AI interpretability must be found in order to develop accurate models that also respect ethical principles in RLS [40].

## 4.6 Remaining Ethical Dilemmas

After conducting an ethical evaluation, we found that there were several ethical dilemmas which require discussion and global consensus. One such ethical dilemma - that has been rarely discussed in literature – is the question of whether ethical boundaries for AI in healthcare should change in drastic, resource-limited scenarios. For example, currently regulations do not allow AI decision support systems to be the final decision-maker, as the algorithms are not yet reliable enough [16]. However, in emergencies and extreme resource shortages, there may not be enough healthcare professionals to handle the situation at hand, even with the support of AI. Thus, due to limited resources, many patients may not be able to access any medical advice or care. Evidently, these resource-limited scenarios are starkly different from the usual conditions of high-income settings, and therefore, the ethical implications must be carefully considered. Should ethical principles, such as the requirement for human oversight, be relaxed in these scenarios to prioritize maximizing the number of lives saved? Or does such a shift in ethical boundaries risk setting a dangerous precedent for AI use in healthcare? Ultimately, these scenarios pose a critical question: Should ethical lines be crossed to increase overall benefit in extreme circumstances, or are there boundaries that must remain unbreakable regardless of the situation? Addressing these questions will require global consensus, interdisciplinary collaboration, and proactive scenario planning to ensure that any adjustments to ethical frameworks maintain both fairness and humanity in deployment of new AI technologies.

Another ethical dilemma centred around maximizing overall benefit is the concern of whether AI technologies truly represent the best allocation of the already scarce resources in RLS. Given that AI-based healthcare tools often require extensive testing and high initial investment, RLS must weigh the value of these technologies against more immediate interventions, such as training additional clinicians and enhancing access to essential medicines [13]. For example, in some rural areas, non-technological interventions such as providing more training and retention programs for on-site medical professionals may be more useful than AI applications [13]. This debate intensifies when one considers that a substantial portion of a constrained budget might be diverted towards AI development and maintenance, leaving fewer resources for more traditional yet cost-effective measures that can reliably deliver tangible health benefits [5]. Furthermore, investing heavily in AI infrastructure may not yield proportional benefits in RLS, especially if these systems remain expensive to maintain and update [41]. There have been few studies regarding the large-scale adoption and deployment of AI in RLS, and so the cost-effectiveness remains yet to be assessed [41]. Thus, these dilemmas raise a pressing another question: does pursuing cutting-edge AI technology truly serve the greatest good, or would allocating funds to strengthen foundational healthcare systems deliver broader and more equitable benefits to communities in the long run?

As discussed earlier, the issue of bias and fairness is a major ethical consideration for AI applications in RLS healthcare systems, potentially leading to violations of fairness and equality of care. A potential strategy that could be used to prevent this is to exclude certain discriminative parameters from the training datasets of AI models. However, it is important to note that the removal of such parameters may significantly lower the model's accuracy and reliability [40]. Thus, a trade-off must be made between individual and societal levels of justice, with the question arising of whether a suitable balance can be found. Researchers, policy-makers, and ethicists must decide whether the potential dangers of bias justify sacrificing model performance, and if so, how much. Furthermore, bias is difficult to quantify, and so methods must be developed in order to detect and measure bias in AI healthcare applications.

## 5 Recommendations for Ethical Implementation

As evident from the evaluation, there are many ethical issues regarding AI health applications in RLS, establishing the need for careful and responsible implementation. The first priority is to develop specific ethical regulations for RLS that recognise their unique healthcare challenges, and infrastructural limitations. Instead of taking a "one size fits all" approach and imposing frameworks designed for high-income and resource-rich settings, frameworks must be context-specific for RLS. In doing so, local regulatory frameworks should align with global standards but maintain enough flexibility

to address region-specific concerns, such as limited data governance and unpredictable financing [2, 5].

Equally crucial is the alignment of AI applications with local cultural values and traditions, ensuring that these technologies strengthen communities rather than disrupt them [13]. The coexistence of diverse ethnic groups, languages, and healthcare practices are common in RLS, amplifying the need for evaluations to be completed on case-by-case basis to accommodate for each unique setting [5, 6]. To prevent the emerging concern of "new technological colonisation", newly deployed AI applications should avoid imposing the paradigms and operational processes of high-income countries on RLS, without proper consideration for local practices and values [5]. Furthermore, to avoid the exploitation of local populations under the guise of innovation, developers and implementers must prioritise fair benefit-sharing, where RLS communities gain tangible returns, rather than serving as data sources or pilot trial sites [13, 41]. To achieve this, it is strongly recommended that local stakeholders - such as community leaders, healthcare professionals, and patients – are given the opportunity to be actively involved in the design and implementation of AI tools. Such stakeholders can provide vital insight into the local health system and cultural specificities, whilst also ensuring that their rights are adequately protected.

Regarding implementation strategies, it is recommended that a gradual and phased introduction of AI technologies in RLS is used, as it allows local communities to adapt without overwhelming existing healthcare structures [2]. Rather than seeking to replace current practices, developers should integrate AI into existing systems, allowing local institutions to acclimatise to new technologies whilst maintaining familiar workflows [2]. To aid RLS in what is a complex and difficult implementation process, high-income nations and resource-rich organisations can be a powerful ally by sharing resources and expertise. Such collaborations can be extremely beneficial, as they can facilitate valuable knowledge exchange, giving RLS an opportunity to learn from the experiences of other AI deployments [13]. In addition to utilising external expertise, it is equally important to concurrently educate the local community about new AI applications, as lower health literacies can hinder acceptance and lead to issues regarding consent and autonomy [44]. Awareness campaigns and educational initiatives should be launched in parallel with new technologies and include objective information regarding benefits and limitations of AI tools. Capacity building initiatives should also be implemented to train regional professionals to operate and maintain AI applications, as they can foster long-term self-reliance and ensure that projects remain viable beyond initial pilot phases [13, 41].

Once AI applications have been deployed, regular audits of output accuracy as well as ethical evaluations of bias, privacy, and autonomy are required to ensure that algorithms remain aligned with local health priorities [13]. Continuous training programs are also crucial, as they allow workers to receive ongoing support for AI systems which will inevitably evolve and update over time [41]. Finally, it is vital to guard against over-reliance on AI tools in RLS healthcare systems. In the case of systems malfunction or connectivity failure, there must be safety net mechanisms in place and healthcare professionals must be able to deliver care regardless of AI assistance.

## 6 Conclusion

While AI holds immense potential to improve healthcare quality in RLS, it simultaneously introduces complex ethical considerations. Currently, there are a wide range of innovative AI tools being developed which can have extremely useful applications in diagnostics, predictive analytics, telemedicine, and resource management. By optimising efficiency, AI can help RLS face shortages of healthcare professionals, higher disease burdens, and fragmented infrastructures. However, these settings remain particularly vulnerable to ethical pitfalls, including algorithmic bias, privacy breaches, reduced patient autonomy, and limited transparency, all of which risk worsening rather than alleviating global disparities. These ethical issues are further complicated by additional dilemmas – such as whether AI should assume decision-making authority during emergencies or whether scarce resources might be more effectively spent on basic health infrastructure. Thus, it is vital to address such concerns through comprehensive planning and careful implementation strategies. The recommendations provided - ranging from gradual implementation to capacity-building measures - offer a roadmap toward fully harnessing AI's capabilities while safeguarding patient welfare and social equity. Local stakeholders should be encouraged to become active contributors, as their input can guide AI solutions to align with community priorities and ensure tangible shared benefits.

Furthermore, future research should focus on addressing the current gaps in quantitative evidence through the development of well-designed, large-scale studies that enable robust meta-analyses and comparative evaluations of AI's impact in RLS. In addition, there is a critical need to establish comprehensive ethical frameworks tailored to the unique challenges of these contexts, addressing algorithmic bias, privacy concerns, and patient autonomy while remaining sensitive to local cultural values and healthcare practices. Ultimately, achieving the full potential of AI in healthcare will require an ongoing commitment to ethical reflection and collaboration among local communities, global stakeholders, and policy-makers. Only by uniting around a shared, ethical approach can AI truly become a catalyst for an equitable and lasting transformation.

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
