# OpenReview forum: "AI in Healthcare for Resource Limited Settings: An Exploration and Ethical Evaluation"
_ACM.org/TheWebConf/2025/Workshop/TIME — TIME 2025 Oral_

### Official Review · Reviewer_ttRK · 2025-01-08
**AI in Healthcare for Resource Limited Settings: An Exploration and Ethical Evaluation**

**Rating:** 4
**Confidence:** 3

**Review:**

1.The works lacks indepth survey- the selection and reason of literature selection is not clear
2.The relvance is not clear

---

### Official Review · Reviewer_7eg6 · 2025-01-12
**Review of AI in Healthcare for Resource Limited Settings: An Exploration and Ethical Evaluation**

**Rating:** 7
**Confidence:** 3

**Review:**

The paper is well-organized, covering applications, ethical evaluations, and actionable recommendations.
It offers a nuanced perspective on the role of AI in RLS, which is a relevant and under explored domain.
The ethical exploration goes beyond the surface, addressing practical and philosophical challenges.
The discussion on global health disparities and the need for culturally sensitive frameworks is impactful.

Global Relevance:
The paper tackles a critical issue that aligns with global health priorities, making it timely and significant.

Rating :
Due to limited technical , economic depth and lake of concrete implementation frameworks keeping it at rating 7

---

### Official Review · Reviewer_sBke · 2025-01-13

**Rating:** 5
**Confidence:** 1

**Review:**

This article raises so many questions that it looks like a survey. I'm not sure if TIME receives survey, but if it does, it looks okay overall. If it doesn't receive it, it should be rejected.

---

### Official Review · Reviewer_F8XJ · 2025-01-13
**Reviews of "AI in Healthcare for Resource Limited Settings: An Exploration and Ethical Evaluation"**

**Rating:** 5
**Confidence:** 4

**Review:**

This paper discusses the potential risks and ethical violations associated with AI in Healthcare for Resource-Limited Settings (RLS).

Advantages:

1. The paper provides a comprehensive discussion of AI healthcare applications for RLS and summarizes relevant ethical issues.
2. It offers recommendations for ethical implementation to promote the responsible use of AI health applications in RLS.

Disadvantages:

1. To my knowledge, existing AGI also have ethical issues. However, this paper focuses solely on AI in healthcare for RLS without linking it to the broader ethical concerns of AGI, which significantly compromises its comprehensiveness.
2. Ethical evaluation should include some quantitative data for support, but this paper only discusses the issues without providing supporting data.
3. The market size and maturity of AI in healthcare for RLS remain unclear.

---

### Official Review · Reviewer_ikEi · 2025-01-20
**Minor improvements but good paper overall.**

**Rating:** 9
**Confidence:** 5

**Review:**

The paper titled "AI in Healthcare for Resource Limited Settings: An Exploration and Ethical Evaluation" presents a thorough examination of the potential applications of artificial intelligence (AI) in healthcare, particularly in resource-limited settings (RLS). It effectively highlights the transformative capabilities of AI while addressing the critical ethical considerations that must accompany its implementation.

However, there are few areas for minor improvements.
1. Firstly, the paper could benefit from a clearer explanation of how studies were weighted and selected. While it mentions using "a range of trials, cross-sectional studies, and systematic and scoping reviews," a more detailed methodology would enhance transparency and reproducibility.
2. Statements like "AI can be used to derive insights for resource management, helping forecast supply needs, optimize the distribution of medical supplies", "There have been few studies regarding the large-scale adoption and deployment of AI in RLS, and so the cost-effectiveness remains yet to be assessed" demonstrates the authors acknowledgment of the current limitations in quantitative synthesis and the need for more robust meta-analytic approaches to strengthen the evidence base for AI applications in resource-limited settings. The quotes highlight that while the current research provides valuable qualitative insights, there is a significant opportunity to enhance the methodological rigor through more comprehensive statistical analysis and meta-analytic techniques.
3. Future research should focus on developing comprehensive ethical guidelines tailored to RLS, addressing issues such as algorithmic bias, privacy concerns, and patient autonomy. There is a pressing need for interdisciplinary collaboration to create adaptive frameworks that respect local cultural values while integrating global ethical standards. By prioritizing these improvements, the paper could significantly contribute to the ongoing discourse on responsible AI deployment in healthcare. The sub section about future considerations can be added in the conclusion which can cover the points on future research.

In conclusion, this paper is of high quality and offers valuable insights into the intersection of AI and healthcare in RLS. Its emphasis on ethical considerations and local engagement is particularly relevant in today's rapidly evolving technological landscape. With some refinements in methodology and expanded research focus, it has the potential to serve as a foundational piece for future studies in this critical area.

---

### Meta-Review · Area_Chair_G2Dd · 2025-01-25

**Recommendation:** Accept (Oral)
**Confidence:** 5

**Metareview:**

The paper dives into AI in healthcare for resource-limited settings. It does a great job in covering a broad range of applications, from decision support to telemedicine, and predictive analytics for disease outbreaks, thoughtfully tying everything together with ethical considerations. That said, the lack of real-world case studies makes some of the arguments feel less grounded.

Clarity and Originality:
The writing is clear and organized, with sections that are easy to follow. The authors did a good job explaining complex ideas, but some technical aspects, like the role of spatiotemporal modeling in predictive analytics, could use simpler language or visuals to improve accessibility. The focus on ethics and culturally sensitive frameworks in AI applications is refreshing and much-needed. The paper brings up some unique points, like involving local stakeholders, but a few suggestions feel more like practical tweaks than groundbreaking solutions.

Pros: The paper highlights real gaps in deploying AI in resource-limited settings, especially around bias and privacy. The recommendations, like gradual implementation and capacity building, are actionable and make sense for the challenges being discussed.
Cons: The heavy reliance on secondary research makes the paper feel a bit theoretical. Adding specific examples or case studies would’ve made the arguments stronger. Also, some sections could dig deeper into the practical side of implementation.

Overall: The paper raises important points and offers thoughtful solutions, making it worth presenting. Its insights would benefit a wider audience.

---

### Decision · Program_Chairs · 2025-01-26

**Decision:**

Accept (Oral)

**Comment:**

The program chair concurs with the area chair's decision.

For the camera-ready version, please revise your paper according to the feedback provided by the reviewers.

Workshop papers must be written in English, follow a double-column format, and comply with the [ACM template](https://www2025.thewebconf.org/short-papers) and formatting guidelines. The template is also available in [Overleaf](https://www.overleaf.com/latex/templates/association-for-computing-machinery-acm-sig-proceedings-template/bmvfhcdnxfty). For authors using Microsoft Word, the Word Interim Template is recommended.

Camera-ready versions of accepted papers can and should include all information to identify authors, and should acknowledge any funding received that directly supported the presented research.

In addition, ensure that the DOI (to be provided by the PCs at a later stage) is included, and cite the workshop (to appear) using the following reference:

```
@inproceedings{time2025,
  title={TIME 2025: 1st International Workshop on Transformative Insights in Multi-faceted Evaluation},
  author={Lei Wang and Md Zakir Hossain and Syed Islam and Tom Gedeon and Sharifa Alghowinem and Isabella Yu and Serena Bono and Xuanying Zhu and Gennie Nguyen and Nur Haldar and Seyed Jalali and Abdur Razzaque and Imran Razzak and Rafiqul Islam and Shahadat Uddin and Naeem Janjua and Aneesh Krishna and Manzur Ashraf},
  booktitle={ACM Web Conference Workshop},
  year={2025}
}
```

Please note that at least one in-person registration is required for each accepted workshop paper to be included in the Companion Proceedings of WWW 2025. All accepted papers must be presented at the conference. Papers not presented (no-shows) may be withdrawn from the companion proceedings. Presentations will be conducted in two formats: oral and poster.

The camera-ready deadline for workshop papers is 7 February 2025 (AoE).